🔓 | **Open Peer Review** | Bacteriology | Research Article

# Six-year retrospective comparison of GeneXpert MTB/RIF Ultra PCR and culture for pediatric tuberculosis diagnosis in non-sputum samples

**Mohammed Suleiman,[1,2] Andrés Pérez-López,[1,3] Ruwa Mohamed,[4] Faheem Mirza,[1] Amina Souli,[1] Humna Shafaq,[1] Muhammad Iqbal[1]**

**ABSTRACT** Diagnosing tuberculosis (TB) in children is challenging due to the paucibacillary nature of the disease and difficulty in obtaining sputum samples. This 6-year retrospective study compared the diagnostic performance of the GeneXpert MTB/RIF Ultra PCR (GX-MTB/R-U) assay, an improved version of the MTB/RIF PCR (GX-MTB/R) assay, with conventional mycobacterial culture in pediatric non-sputum samples. A total of 417 non-sputum samples from a tertiary pediatric facility underwent simultaneous GX-MTB/R-U and culture testing. Performance was evaluated using conventional culture as a gold standard, and clinical review was used to resolve discordant results. GX-MTB/R-U demonstrated 93.9% sensitivity and 97.9% specificity when culture was used as the gold standard. Eight samples positive by GX-MTB/R-U assay but negative by culture were confirmed as true positive using clinical review of the patients, increasing the sensitivity to 95.1% and specificity to 100%. GX-MTB/R-U demonstrated excellent diagnostic performance in comparison to conventional culture and microscopy, particularly for paucibacillary and extrapulmonary samples. Our study results support the use of GX-MTB/R-U assay as a frontline diagnostic tool in pediatric TB, with culture reserved for confirmation of negative PCR results and antibiotic susceptibility testing.

**IMPORTANCE** Delays or inaccuracies in tuberculosis (TB) diagnosis can lead to treatment delays, disease progression, and poor clinical outcomes. This study provides compelling evidence on the excellent diagnostic performance of the GeneXpert MTB/RIF Ultra (GX-MTB/R-U) assay in pediatric non-sputum samples compared to conventional culture, the current gold standard. By demonstrating high sensitivity and specificity, particularly in extrapulmonary samples, the findings of our study highlight the potential of GX-MTB/R-U as a frontline TB diagnostic tool. The use of this assay in clinical practice could improve early detection, ensure timely initiation of antibiotic treatment, ultimately improving pediatric TB management and outcomes.

**KEYWORDS** TB PCR, tuberculosis, pediatric, TB diagnosis

Tuberculosis (TB) continues to be a major cause of morbidity and mortality worldwide, with an estimated 1.3 million new cases diagnosed in 2022 among children aged 0–14, including 115,000 cases caused by multidrug-resistant strains (1). Early and accurate diagnosis is critical but challenging in this population due to the typically paucibacillary nature of the disease, non-specific clinical presentations, and the inability of young children to expectorate sputum (2, 3). Therefore, it is essential to enhance the diagnostic yield of TB in children from non-sputum specimens such as gastric aspirates, bronchoalveolar lavage (BAL), cerebrospinal fluid (CSF), lymph nodes, bone, tissues, and other extrapulmonary samples (4). Although extrapulmonary TB (EPTB)

**Peer Reviewer** Robert Matovu, Makerere University, Kampala, Uganda

Address correspondence to Mohammed Suleiman, MSuleiman@sidra.org.

The authors declare no conflict of interest.

represents a substantial proportion of cases in the pediatric population, mycobacterial culture, the current diagnostic gold standard, has notable limitations in this setting (5–7). These include the long turnaround times (up to 42 days), reduced sensitivity, the need for specialized training, biosafety concerns, and complex laboratory infrastructure, which may contribute to underdiagnosis or delayed treatment initiation (5–7). Due to these limitations and despite the recent advances in the laboratory diagnosis of TB, clinical symptoms, chest radiography, and contact history remain the cornerstone of TB diagnosis (8, 9).

The introduction of rapid molecular diagnostics, especially the GX-MTB/R-U assay (Sunnyvale, CA, USA), endorsed by the World Health Organization (WHO), has significantly advanced TB detection by providing faster, more sensitive identification of *Mycobacterium tuberculosis* complex and rifampicin resistance simultaneously from sputum samples in less than 2 h (7, 10). Most large-scale studies directly comparing the performance of the GX-MTB/R-U assay against traditional culture in non-sputum samples were conducted on adult patients, with limited data available on the pediatric population (11, 12). The few studies performed in the pediatric population assessed the performance of the GX-MTB/R-U assay on sputum samples, which is the FDA-approved specimen type for this assay (6, 13–17).

This retrospective study aims to address this gap by evaluating the diagnostic performance of the GX-MTB/R-U assay versus conventional mycobacterial culture in 417 non-sputum samples collected over 6 years in a tertiary pediatric healthcare setting in Qatar.

## MATERIALS AND METHODS

### Study design and setting

This retrospective study included all non-sputum samples submitted for GX-MTB/R-U testing to our microbiology laboratory at Sidra Medicine between January 2019 and December 2024 from patients aged 0–18. Sidra Medicine is a pediatric tertiary care hospital serving as the main pediatric subspecialty referral center in the State of Qatar. All samples were collected in a sterile container and tested immediately upon receipt in our laboratory.

### Diagnostic tests

All samples were tested using the GX-MTB/R-U assay (Cepheid, Sunnyvale, CA, USA) in the microbiology laboratory at Sidra Medicine and referred to the reference laboratory for TB culture. The TB culture included acid-fast bacilli (AFB) smear, liquid media using the BACTEC MGIT 960 (Becton Dickinson, Sparks, MD, USA) supplemented with OADC enrichment and PANTA antibiotic mixture, and solid media using Lowenstein-Jensen slants. Cultures were incubated for 42 days. For the GX-MTB/R-U assay, all samples were processed in the same way as sputum samples, as described in the package insert (Cepheid, Sunnyvale, CA, USA) (13). Tissue and biopsy samples were mechanically homogenized using a tissue grinder before processing. Sterile body fluids were centrifuged at $5,000 \times g$ for 15 min at room temperature, then the pellet was resuspended in the sample reagent provided in the kit in a 2:1 ratio before processing. All results were interpreted according to the manufacturer's instructions.

### Statistical analysis

The performance of the GX-MTB/R-U assay was assessed by comparing the test result against the gold standard TB culture results. Overall and sample type-specific sensitivity, specificity, positive predictive value (PPV), and negative predictive value (NPV) for GX-MTB/R-U assay were calculated. The 95% confidence intervals were calculated using the Wilson score method. Agreement between assays was assessed using positive percent agreement (PPA), negative percent agreement (NPA), and Cohen's kappa

coefficient. Statistical calculations were performed using the online version of Med-Calc (https://www.medcalc.org/calc/). Discordant results were resolved by performing a clinical review by an infectious disease physician. The clinical review included reviews of clinical symptoms such as fever, cough, and weight loss, radiological findings, other laboratory investigations, travel to TB-endemic regions, response to antibiotic treatment, history of Bacillus Calmette-Guérin (BCG) vaccination, and history of contact with TB cases. Patients with clinical presentations consistent with TB upon review were classified as true positives for TB.

## RESULTS

### Agreement between GX-MTB/R-U assay and culture

A total of 417 samples were included in this study from 296 unique patients. Seventy-nine patients had two or more samples. One hundred fifty-three (51.7%) of the patients included in this study were male, and the median age was 6 years (interquartile range [2–11]). Tables 1 and 2 show the breakdown of samples according to specimen type. Out of the 417 samples, 407 (97.6%) had fully concordant results between the GX-MTB/R-U assay and culture. The GX-MTB/R-U assay yielded 39/417 (9.3%) positive results, while culture yielded 31/417 (7.4%) positive results. Thirty-one samples were positive by both methods with an overall 93.9% PPA. Three hundred seventy-six samples were negative by both methods with an overall 97.9% NPA. The two methods used in our study had strong agreement, Cohen's $\kappa$ = 0.84 (0.76–0.94), with only 10 samples showing discordant results. Two gastric aspirates collected from the same patient, which were positive by culture but negative by the GX-MTB/R-U assay, were confirmed as true positives by the clinical review (Table 3). On the other hand, the eight samples from eight different patients (four tissues-biopsies, two lymph nodes, one abscess, and one tracheal aspirate), which were negative by culture but positive by the GX-MTB/R-U assay, were confirmed as true positives by the clinical review (Table 3).

### Performance characteristics of the GX-MTB/R-U assay

The sensitivity, specificity, PPV, and NPV of the GX-MTB/R-U assay against the results of the gold standard culture are shown in Table 1. The sensitivity, specificity, PPV, and NPV of the GX-MTB/R-U assay after resolving discrepant results using clinical review of patient charts are shown in Table 2. Overall, in comparison to the gold standard

**TABLE 1** GX-MTB/R-U assay performance against culture[a]

| Specimen type | Number of samples | TP | TN | FN | FP | % TP | Sensitivity % (95% CI) | Specificity % (95% CI) | PPV % (95% CI) | NPV % (95% CI) |
|---|---|---|---|---|---|---|---|---|---|---|
| BAL | 139 | 1 | 138 | 0 | 0 | 0.7 | 100 (2.5–100) | 100 (97.4–100) | 100 (2.5–100) | 100 (97.4–100) |
| Gastric aspirate | 75 | 9[b] | 64 | 2[c] | 0 | 12 | 81.8 (48.2–97.7) | 100 (94.4–100) | 100 (66.4–100) | 97 (90.1–99.1) |
| Tissue-biopsy | 72 | 13 | 55 | 0 | 4 | 18.1 | 100 (75.3–100) | 93.2 (83.5–98.1) | 76.5 (55.8–89.3) | 100 (93.5–100) |
| Abscess | 34 | 6[b] | 27 | 0 | 1 | 17.6 | 100 (54–100) | 96.4 (81.7–99.9) | 85.7 (46.7–97.6) | 100 (87.2–100) |
| CSF | 23 | 0 | 23 | 0 | 0 | 0 | NA | 100 (85.2–100) | NA | 100 (85.2–100) |
| Pleural fluid | 21 | 1 | 20 | 0 | 0 | 4.8 | 100 (2.5–100) | 100 (83.2–100) | 100 (2.5–100) | 100 (83.2–100) |
| Other sterile body fluid[d] | 18 | 0 | 18 | 0 | 0 | 0 | NA | 100 (81.5–100) | NA | 100 (81.5–100) |
| Bone marrow | 12 | 0 | 12 | 0 | 0 | 0 | NA | 100 (81.5–100) | NA | 100 (81.5–100) |
| Wound | 9 | 1 | 8 | 0 | 0 | 11.1 | 100 (2.5–100) | 100 (63.1–100) | 100 (2.5–100) | 100 (63.1–100) |
| Tracheal aspirate | 7 | 0 | 6 | 0 | 1 | 0 | NA | 85.7 (42.1–99.6) | NA | 100 (54–100) |
| Lymph node | 7 | 0 | 5 | 0 | 2 | 0 | NA | 71.4 (29–96.3) | NA | 100 (47.8–100) |
| Total | 417 | 31 | 376 | 2 | 8 | 7.4 | 93.9 (79.8–99.3) | 97.9 (95.9–99.1) | 79.5 (66–88.5) | 99.5 (98–99.9) |

[a]BAL, bronchoalveolar lavage; CI, confidence interval; CSF, cerebrospinal fluid; FN, false negative; FP, false positive; NA, not applicable; NPV, negative predictive value; PPV, positive predictive value; TN, true negative; TP, true positive.
[b]All positive samples are from a single patient with the exception of gastric aspirate (nine samples from five patients) and abscess (six samples from five patients).
[c]Two FN samples from the same patient.
[d]Other sterile body fluid: 3 ascitic fluids, 1 abdominal fluid, 1 pancreatic fluid, 4 pericardial fluids, 5 synovial fluids, and 4 peritoneal fluids.

**TABLE 2** GX-MTB/R-U assay performance after resolving discordant results[a]

| Specimen type | Number of samples | TP | TN | FN | FP | % TP | Sensitivity % (95% CI) | Specificity % (95% CI) | PPV % (95% CI) | NPV % (95% CI) |
|---|---|---|---|---|---|---|---|---|---|---|
| BAL | 139 | 1 | 138 | 0 | 0 | 0.7 | 100 (2.5–100) | 100 (97.4–100) | 100 (2.5–100) | 100 (97.4–100) |
| Gastric aspirate | 75 | 9[b] | 64 | 2[c] | 0 | 12 | 81.8 (48.2–97.7) | 100 (94.4–100) | 100 (66.4–100) | 97 (90.1–99.1) |
| Tissue-biopsy | 72 | 17 | 55 | 0 | 0 | 23.6 | 100 (80.5–100) | 100 (93.5–100) | 100 (80.1–100) | 100 (95–100) |
| Abscess | 34 | 7[b] | 27 | 0 | 0 | 20.6 | 100 (59–100) | 100 (87.2–100) | 100 (59–100) | 100 (87.2–100) |
| CSF | 23 | 0 | 23 | 0 | 0 | 0 | NA | 100 (85.2-100) | NA | 100 (85.2-100) |
| Pleural fluid | 21 | 1 | 20 | 0 | 0 | 4.8 | 100 (2.5–100) | 100 (83.2–100) | 100 (2.5–100) | 100 (83.2–100) |
| Other sterile body fluid[d] | 18 | 0 | 18 | 0 | 0 | 0 | NA | 100 (81.5–100) | NA | 100 (81.5–100) |
| Bone marrow | 12 | 0 | 12 | 0 | 0 | 0 | NA | 100 (81.5–100) | NA | 100 (81.5–100) |
| Wound | 9 | 1 | 8 | 0 | 0 | 11.1 | 100 (2.5-100) | 100 (63.1–100) | 100 (2.5-100) | 100 (63.1–100) |
| Tracheal aspirate | 7 | 1 | 6 | 0 | 0 | 14.2 | 100 (2.5–100) | 100 (54–100) | 100 (2.5–100) | 100 (54–100) |
| Lymph node | 7 | 2 | 5 | 0 | 0 | 28.5 | 100 (15.8–100) | 100 (47.8–100) | 100 (15.8–100) | 100 (47.8–100) |
| Total | 417 | 39 | 376 | 2 | 0 | 9.4 | 95.1 (83.5–99.4) | 100 (99–100) | 100 (90.1–100) | 99.5 (98–99.9) |

[a]BAL, bronchoalveolar lavage; CI, confidence interval; CSF, cerebrospinal fluid; FN, false negative; FP, false positive; NA, not applicable; NPV, negative predictive value; PPV, positive predictive value; TN, true negative; TP, true positive.
[b]All positive samples are from a single patient with the exception of gastric aspirate (nine samples from five patients) and abscess (seven samples from six patients).
[c]Two FN samples from the same patient.
[d]Other sterile body fluid: 3 ascitic fluids, 1 abdominal fluid, 1 pancreatic fluid, 4 pericardial fluids, 5 synovial fluids, and 4 peritoneal fluids.

culture, the GX-MTB/R-U assay showed excellent performance with 93.9% sensitivity, 97.9% specificity, 79.5% PPV, and 99.5% NPV. The performance of the GX-MTB/R-U assay after the clinical review of discordant results was further improved to 95.1% sensitivity, 100% specificity, 100% PPV, and 99.5% NPV. The breakdown of sample types is shown in Tables 1 and 2. The sample types with the highest percentage of positive GX-MTB/R-U assay were lymph nodes with 28.5% and tissue-biopsies with 23.6% after resolving discordant results.

## Rifampicin resistance detected by GX-MTB/R-U assay

Among the 39 samples with positive GX-MTB/R-U assay results, only two samples (5.2%) were detected to be rifampicin resistant. Among all samples tested, the resistance rate is 2/417 (0.5%). These two results were confirmed by antibiotic susceptibility testing performed on the BACTEC MGIT 960 from the corresponding positive cultures.

## AFB smear results

Of the 417 samples studied, AFB smear was performed on 415 (99.5%), with 16 (3.9%) yielding positive results. In comparison to culture, the AFB smear exhibited poor sensitivity (40.6%) and excellent specificity (99.2%). Similarly, after clinical review, the smear exhibited poor sensitivity (40.0%) and excellent specificity (100%). All 16 positive smear samples had positive GX-MTB/R-U assay results (100% PPA), and 13/16 positive smears had positive culture (81.2%), which were later confirmed as true positives after clinical review. The AFB smear results were falsely negative in 16 samples that were positive by both culture and GX-MTB/R-U assay.

## DISCUSSION

The diagnosis of TB in children is challenging because the disease is often paucibacillary, sputum specimens are difficult to obtain, and the long turnaround time of traditional TB cultures (2, 3, 5–7). Of particular concern is the higher proportion of EPTB in which culture yield is limited, and the higher risk of progression to severe forms of disease, particularly TB meningitis and miliary TB in children, especially those younger than 3 years of age (<3 years old) (9, 18). The focus of our study is to find a reliable and quicker diagnostic test for TB in children using non-sputum samples.

The GX-MTB/R-U assay has several advantages over traditional cultures, including excellent sensitivity and specificity across varying specimen sites from pediatric patients,

**TABLE 3** Results of clinical review of discordant samples[a]

| Discordant sample number | Specimen type | GX-MTB/R-U assay result | Smear result | Culture result | Clinical diagnosis | Treatment | Significant findings from clinical review |
|---|---|---|---|---|---|---|---|
| 1 | Tissue-biopsy | Positive | Negative | Negative | BCG ulcer | Yes | 1 year old, suspected immunodeficiency, 6 months history of non-resolving left thigh wound collection at BCG vaccine site. |
| 2 | Tissue-biopsy | Positive | Negative | Negative | BCG lymphadenopathy | No | 1-year-old HIV patient, right axillary swelling at BCG vaccine site. Histopathology results consistent with granulomatous lymphadenitis (necrotizing). |
| 3 | Tissue-biopsy | Positive | Negative | Negative | Disseminated TB | Yes | 17 years old, cough, weight loss, right side pleural effusion, chronic abdominal pain. Abdomen CT scan showed multiple lymph nodes. Histopathology results consistent with necrotizing granulomatous inflammation (peritoneal). |
| 4 | Lymph node | Positive | Positive | Negative | TB lymphadenopathy | Yes | 13 years old, fever, weight loss, multiple lymph node swelling (supraclavicular, cervical, elbow, paratracheal). Histopathology results consistent with granulomatous necrotizing lymphadenitis, CT chest showed multiple calcified mediastinal lymph nodes. |
| 5 | Tissue-biopsy | Positive | Negative | Negative | BCG lymphadenopathy | No | 4 months old, left axillary swelling at BCG vaccine site. Histopathology results consistent with necrotizing granulomatous inflammation. |
| 6 | Abscess | Positive | Positive | Negative | BCG lymphadenopathy | No | 5 months old, history of left neck and supraclavicular swelling. |
| 7 | Tracheal aspirate | Positive | Negative | Negative | Pulmonary TB | Yes | 3 months old, immunodeficiency, positive GX-MTB/R-U assay from respiratory sample. Chest X-ray and CT scan consistent with TB. |
| 8 | Lymph node | Positive | Positive | Negative | BCG lymphadenopathy | No | 14 months old, left axillary swelling at BCG vaccine site. Histopathology results consistent with necrotizing granulomatous inflammation. |
| 9 and 10 | Gastric aspirate | Negative | Negative | Positive | Pulmonary TB | Yes | 9 years old, cough for 1 month, weight loss, radiological changes in chest X-ray and CT scan. Interferon Gamma Release Assay positive. |

[a]BCG, Bacillus Calmette-Guérin; CT, computed tomography; HIV, human immunodeficiency virus.

as shown in the results of this study. A smaller study performed in India, which included 171 extrapulmonary samples, revealed comparable results with the GX-MTB/R-U assay showing 88.9% sensitivity and 98% specificity in comparison to culture (19). It is noteworthy that our results were superior to the results of a systematic review and meta-analysis, which looked at the accuracy of the GX-MTB/R assay in diagnosing EPTB in pediatric patients across eight studies (20). This systematic review revealed that the GX-MTB/R assay has a moderate pooled sensitivity of 71% (95% CI 0.63–0.79) and excellent pooled specificity of 97% (95% CI 0.95–0.99), across the 652 pediatric samples included in the eight studies (20). It is plausible that the higher sensitivity in our study may be attributed to the use of the newer ultra-version of the GX-MTB/R assay which was introduced in 2018 and has demonstrated better performance than the previous version (21). According to the manufacturer's data, the estimated limit of detection (LOD) for MTB (H37Rv strain) for the newer ultra version of the assay is 11.8 CFU/mL in comparison to the older version of this assay with an estimated LOD at 414 CFU/mL (13,

22). A systematic review, which included 5,855 samples across 19 studies, showed that the GX-MTB/R-U assay was consistently more sensitive (pooled sensitivity: 84%) than the previous version of the GX-MTB/R assay (pooled sensitivity: 69%) (21).

In our study, only gastric aspirates showed lower sensitivity at 81.8%. This is consistent with two previous studies performed in China on gastric aspirates, and they reported sensitivity of the GX-MTB/R-U assay at 85%–87% (9, 23). The consolidated TB guidelines from WHO have highlighted that all current TB diagnostic tests are suboptimal in diagnosing TB in children from gastric aspirates due to the paucibacillary nature of the disease in the pediatric population (24). Therefore, a positive result accurately diagnoses a case of TB, while a negative result does not exclude the diagnosis of TB disease (24).

After resolving discordant samples, the GX-MTB/R-U assay detected eight more positive samples, including lymph nodes and tissue biopsies, which showed the highest yield among positive specimens in our study, 28.5% and 23.6%, respectively. This highlights the advantage of using this GX-MTB/R-U assay to diagnose TB in such valuable and precious samples in an accurate and quick manner. In a study performed on 109 biopsy samples, the GX-MTB/R assay was positive in 17 (15.6%) with 73.9% sensitivity and 100% specificity (25). As mentioned earlier, the higher positivity in our study could be attributed to the use of the ultra-version of the assay which showed higher sensitivity and improved LOD (21).

Another advantage of the GX-MTB/R-U assay is the faster reporting time of less than 2 h in comparison to smear, which is normally processed during routine working hours, and the results are usually not available within the first 24 h after specimens' collection or culture, which can take up to 42 days until final report. Because of the retrospective nature of this study, we were not able to assess the impact of the GX-MTB/R-U assay on clinical management. A recent systematic review, which looked at 45 studies, found that the GX-MTB/R assay on average reduced TB diagnostic delays by 1.79 days and treatment initiation delays by 2.55 days in comparison to TB smear (26).

Additionally, the GX-MTB/R-U assay showed superior sensitivity in comparison to AFB smear and was able to detect 16 positive cases that were missed by AFB microscopy. This is consistent with previous studies which found that the GX-MTB/R-U assay has much superior sensitivity than AFB smear in non-sputum samples (9, 27, 28). Among 129 children with active TB, the sensitivity of the GX-MTB/R-U assay on gastric aspirate was 60.5% in comparison to 10.9% sensitivity of the AFB smear (9). In a study that evaluated CSF samples among 187 children, GX-MTB/R-U assay produced higher sensitivity when compared with AFB smear (88% versus 2%) (27). In another study that evaluated 106 lymph nodes, GX-MTB/R-U assay showed higher sensitivity when compared with AFB smear (75.7% versus 5.4%) (28). Although our study could not assess the accuracy of the GX-MTB/R-U to detect rifampicin resistance due to low prevalence of rifampicin resistance (0.5%) detected among our positive samples, which otherwise concurs with rifampicin resistance rates reported in Qatar (29), the faster detection of rifampicin resistance compared with culture-based methods in EPTB specimens is another potential advantage of the assay, particularly in settings with high prevalence of both isoniazid and rifampicin resistance. A previous systematic review study conducted in China, which assessed 3,801 patients across 11 studies, revealed an excellent diagnostic performance of rifampicin resistance detection in pediatrics with a pooled sensitivity of 94% and a pooled specificity of 99% (30).

Despite all the advantages of using the GX-MTB/R-U assay as a frontline TB diagnostic, this test is not a complete solution for all sample types as it has weaker evidence in children and cannot completely replace culture according to the WHO (10, 24). Culture is still recommended for sample types with insufficient supporting evidence, and for susceptibility testing, especially when suspecting multidrug-resistant TB isolates (10, 24). Furthermore, the WHO emphasizes that a negative molecular test on extrapulmonary specimens does not rule out TB in children; therefore, culture confirmation is still needed in these scenarios (10, 24).

Our study had several limitations. The retrospective nature of our study limited our ability to collect sufficient clinical details for all patients; therefore, the study only focused on the diagnostic performance of the GX-MTB/R-U assay against traditional culture. A clinical review was only performed on discordant samples to confirm if they had a confirmed TB diagnosis based on their clinical history and other diagnostic methods performed such as radiological and laboratory tests. Another limitation is the small number of specific sample types included in our study, particularly CSF and lymph node tissues, and the absence of positive GX-MTB/R-U assay in several specimen types included in this study. This has limited our statistical power for subgroup analysis and performance characteristics such as sensitivity could not be calculated for specific sample types as we did not detect any true positives. However, few previous studies evaluated some of these specific sample types and showed that the GX-MTB/R-U assay performs as well or better than culture. A systematic review of 646 lymph node samples across eight studies from pediatric patients showed moderate sensitivity of 79% and high specificity of 90% for the GX-MTB/R-U assay in comparison to culture (31). Another recent study evaluating the performance of the GX-MTB/R-U assay on CSF samples among 187 children showed a significantly better sensitivity of 88% in comparison to the GX-MTB/R assay and culture that showed sensitivities of 34% and 30%, respectively (27).

In conclusion, our study showed that the GX-MTB/R-U assay is a reliable and rapid method for the diagnosis of EPTB in children. In addition, this method is a valuable diagnostic tool for young children with suspected pulmonary TB who cannot produce sputum. Larger studies are needed to assess the performance of the GX-MTB/R-U assay in extrapulmonary samples, specifically sterile samples such as CSF, lymph nodes, and sterile body sites.

## AUTHOR AFFILIATIONS

[1]Department of Pathology, Sidra Medicine, Doha, Qatar
[2]Department of Biomedical Sciences, Qatar University, Doha, Qatar
[3]Department of Pathology and Laboratory Medicine, Weill Cornell Medicine in Qatar, Doha, Qatar
[4]Department of Infectious Diseases, Sidra Medicine, Doha, Qatar

## AUTHOR ORCIDs

Mohammed Suleiman http://orcid.org/0000-0001-5277-0531
Andrés Pérez-López https://orcid.org/0000-0002-2301-470X

## AUTHOR CONTRIBUTIONS

Mohammed Suleiman, Conceptualization, Formal analysis, Investigation, Supervision, Writing – original draft, Writing – review and editing | Andrés Pérez-López, Conceptualization, Formal analysis, Investigation, Methodology, Supervision, Writing – review and editing | Ruwa Mohamed, Formal analysis, Investigation, Writing – review and editing | Faheem Mirza, Data curation, Formal analysis, Investigation, Validation, Writing – review and editing | Amina Souli, Data curation, Formal analysis, Validation, Writing – review and editing | Humna Shafaq, Data curation, Formal analysis, Validation, Writing – review and editing | Muhammad Iqbal, Conceptualization, Formal analysis, Supervision, Writing – original draft, Writing – review and editing

## DATA AVAILABILITY

The data sets generated and/or analyzed during the current study are available from the corresponding author on reasonable request.

## ETHICS APPROVAL

The study was reviewed and approved by the Institutional Review Board (IRB) at Sidra Medicine (2356050-1).

## ADDITIONAL FILES

The following material is available online.

### Open Peer Review

**PEER REVIEW HISTORY (review-history.pdf).** An accounting of the reviewer comments and feedback.

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
