## [Reviewer comments · Microbiology Spectrum]

Microbiology Spectrum

Six-Year Retrospective Comparison of GeneXpert MTB/RIF Ultra PCR and Culture for Pediatric Tuberculosis Diagnosis in Non-Sputum Samples

Mohammed Suleiman, Andrés Pérez-López, Ruwa Mohamed, Faheem Mirza, Amina Souli, Humna Shafaq, and Muhammad Iqbal

Corresponding Author(s): Mohammed Suleiman, Sidra Medicine

Review Timeline:

Submission Date:	October 1, 2025
Editorial Decision:	October 20, 2025
Revision Received:	November 10, 2025
Editorial Decision:	November 20, 2025
Revision Received:	November 23, 2025
Accepted:	December 2, 2025

Editor: Siu-Kei Chow

Reviewer(s): Disclosure of reviewer identity is with reference to reviewer comments included in decision letter(s). The following individuals involved in review of your submission have agreed to reveal their identity: Robert Matovu (Reviewer #1)

Transaction Report:

DOI: <https://doi.org/10.1128/spectrum.03135-25>

Re: Spectrum03135-25 (**Six-Year Retrospective Comparison of GeneXpert MTB/RIF Ultra PCR and Culture for Pediatric Tuberculosis Diagnosis in Non-Sputum Samples**)

Dear Dr. Mohammed Suleiman:

Thank you for the privilege of reviewing your work. Below you will find my comments, instructions from the Spectrum editorial office, and the reviewer comments.

Revision Guidelines

Sincerely,
Siu-Kei Chow
Editor
Microbiology Spectrum

Reviewer #1 (Comments for the Author):

This paper presents a well-executed study with clear objectives and a good experimental design. The methodology is good, and data are presented clearly and thoroughly, demonstrating careful attention to detail and reproducibility. The authors have employed appropriate control samples and their analytical techniques are suitable for the research questions posed. The study findings are realistically supported by the data, and the discussion well interprets these findings within the context of the current literature. The write up is clear and concise, making the study accessible to readers in the field

Reviewer #2 (Comments for the Author):

The authors perform a six year retrospective review of the performance of the MTB/RIF ultrasensitive assay compared to culture on non-sputum samples in children 18 years old and younger.

Major points:

The number of specimens for each sample type is unclear in the results section. Additional description in the results section would be appropriate. Alternatively, if the authors reference Table 1 earlier in the results section, this may be sufficient. It is also important to note that several specimen types did not have a positive MTB NAAT result in the study. At the very least, this should be addressed in the limitations section.

Is there demographic data available for patients reviewed in the study (M/F, age, etc)? Were there multiple specimens in the study from the same patient? Or does this study include a single positive specimen from individual patients?

Could the authors clarify how clinical adjudication was performed? Other imaging, travel history, laboratory results? Please include a description in the methods section.

Line 112 indicates 8 tissue biopsies as negative by culture but positive by the MTB NAAT. However, comparison of table 1 vs table 2 only shows 4 tissue biopsies, 1 abscess and 1 tracheal aspirate as initial FP MTB NAAT prior to clinical adjudication. Perhaps a table of key clinical and lab results would be helpful for the MTB NAAT positive, culture negative results that were reviewed.

Information on primary specimen AFB stain results is not discussed. If available, it would be helpful addition to the MTB NAAT and culture results. Typically, MTB NAAT have higher positivity rates from specimens that are smear positive and decreased sensitivity on smear negative samples. This reviewer recognizes that some sterile fluid like CSF, bone marrow, etc, may not have a primary smear performed but it might be useful to know on other specimen types when available.

Minor points:

Please include more specific AFB culture details. Which type/s of solid media were used and how long was culture incubation performed. Was liquid media supplemented appropriately?

Include limit of detection for ultrasensitive vs "standard assay" in methods or discussion section.

Line 87, 88 - list manufacturer and headquarters for MTB NAAT

What types of specimens are considered "other sterile body fluids" in table 1 and 2. Please include in methods or as an additional footnote.

Date: 19/10/2025

The journal of Microbiology Spectrum

Name of the Reviewer: Robert Matovu

Manuscript Number: Spectrum03135-25

Title of the Manuscript: Six-Year Retrospective Comparison of GeneXpert MTB/RIF Ultra PCR and Culture for Pediatric Tuberculosis Diagnosis in Non-Sputum Samples

General Comments

This paper presents a well-executed study with clear objectives and a good experimental design. The methodology is good, and data are presented clearly and thoroughly, demonstrating careful attention to detail and reproductivity. The authors have employed appropriate control samples and their analytical techniques are suitable for the research questions posed.

The study findings are realistically supported by the data, and the discussion well interprets these findings within the context of the current literature. The write up is clear and concise, making the study accessible to readers in the field

END

Re: Spectrum03135-25 (Six-Year Retrospective Comparison of GeneXpert MTB/RIF Ultra PCR and Culture for Pediatric Tuberculosis Diagnosis in Non-Sputum Samples)

We would like to thank the editor and reviewers for taking the time to review this work and to provide constructive suggestions to improve the quality of our manuscript. Below, please find our point-by-point responses to the reviewers' comments:

Reviewer 1 comments	Author's response
This paper presents a well-executed study with clear objectives and a good experimental design. The methodology is good, and data are presented clearly and thoroughly, demonstrating careful attention to detail and reproducibility. The authors have employed appropriate control samples, and their analytical techniques are suitable for the research questions posed. The study findings are realistically supported by the data, and the discussion well interprets these findings within the context of the current literature. The write up is clear and concise, making the study accessible to readers in the field	Thank you so much for your comments and feedback

Reviewer 2 comments	Author's response
The number of specimens for each sample type is unclear in the results section. Additional description in the results section would be appropriate. Alternatively, if the authors reference Table 1 earlier in the results section, this may be sufficient. It is also important to note that several specimen types did not have a positive MTB NAAT result in the study. At the very least, this should be addressed in the limitations section.	A reference to table 1 added early in the result section. Limitation section updated according to the reviewer's comment.
Is there demographic data available for patients reviewed in the study (M/F, age, etc)? Were there multiple specimens in the study from the same patient? Or does this study include a single positive specimen from individual patients?	Available demographic data were added. We also clarified that this study included multiple samples from the same patients in the result section including details on patients who had multiple positive samples (highlighted in table 1 and 2).
Could the authors clarify how clinical adjudication was performed? Other imaging, travel history, laboratory	Added to the method section and Table 3.

results? Please include a description in the methods section.	
Line 112 indicates 8 tissue biopsies as negative by culture but positive by the MTB NAAT. However, comparison of table 1 vs table 2 only shows 4 tissue biopsies, 1 abscess and 1 tracheal aspirate as initial FP MTB NAAT prior to clinical adjudication. Perhaps a table of key clinical and lab results would be helpful for the MTB NAAT positive, culture negative results that were reviewed.	Apologize for the oversight, the text has been corrected (4 tissue biopsies, 2 lymph nodes, 1 abscess, and 1 tracheal aspirate). Table 3 added describing the significant results obtained from the clinical review.
Information on primary specimen AFB stain results is not discussed. If available, it would be helpful addition to the MTB NAAT and culture results. Typically, MTB NAAT have higher positivity rates from specimens that are smear positive and decreased sensitivity on smear negative samples. This reviewer recognizes that some sterile fluid like CSF, bone marrow, etc, may not have a primary smear performed but it might be useful to know on other specimen types when available.	Thank you for the suggestion, information about AFB stain results is now added to the manuscript. Results and discussion section updated.
Please include more specific AFB culture details. Which type/s of solid media were used and how long was culture incubation performed. Was liquid media supplemented appropriately	Requested information added in the methods section.
Include limit of detection for ultrasensitive vs "standard assay" in methods or discussion section.	The LOD for both assays added in the discussion.
Line 87, 88 - list manufacturer and headquarters for MTB NAAT	Added in line 88.
What types of specimens are considered "other sterile body fluids" in table 1 and 2. Please include in methods or as an additional footnote.	Added as a footnote to table 1 and 2.

Re: Spectrum03135-25R1 (**Six-Year Retrospective Comparison of GeneXpert MTB/RIF Ultra PCR and Culture for Pediatric Tuberculosis Diagnosis in Non-Sputum Samples**)

Dear Dr. Mohammed Suleiman:

Thank you for the privilege of reviewing your work. Below you will find my comments, instructions from the Spectrum editorial office, and the reviewer comments.

Revision Guidelines

Sincerely,
Siu-Kei Chow
Editor
Microbiology Spectrum

Reviewer #2 (Comments for the Author):

The authors have addressed all of my major comments.

Minor edits suggested for reference to GXMTB/R-U assay for consistency. "GXMTB/R-U assay" is used in most of the manuscript but some areas it is referred to as:
"GeneXpert MTB/RIF assay" - Tables 1 and 2. This should indicate the Ultra assay was used to avoid confusion since the ultra assay has a lower LOD than the standard MTB/RIF

"TB PCR" (line 78, 234)

"PCR" (line 138, 146-149)

Re: Spectrum03135-25R1 (Six-Year Retrospective Comparison of GeneXpert MTB/RIF Ultra PCR and Culture for Pediatric Tuberculosis Diagnosis in Non-Sputum Samples)

We would like to thank the editor and reviewer for taking the time to review this work and to provide constructive suggestions to improve the quality of our manuscript. Below, please find our point-by-point responses to the reviewers' comments:

Reviewer 2 comments	Author's response
Minor edits suggested for reference to GXMTB/R-U assay for consistency. "GXMTB/R-U assay" is used in most of the manuscript but some areas it is referred to as: "GeneXpert MTB/RIF assay" - Tables 1 and 2. This should indicate the Ultra assay was used to avoid confusion since the ultra assay has a lower LOD than the standard MTB/RIF "TB PCR" (line 78, 234) "PCR" (line 138, 146-149)	Thank you for your review and comments. The manuscript has been reviewed and corrected.

Re: Spectrum03135-25R2 (**Six-Year Retrospective Comparison of GeneXpert MTB/RIF Ultra PCR and Culture for Pediatric Tuberculosis Diagnosis in Non-Sputum Samples**)

Dear Dr. Mohammed Suleiman:

Your manuscript has been accepted, and I am forwarding it to the ASM production staff for publication. Your paper will first be checked to make sure all elements meet the technical requirements. ASM staff will contact you if anything needs to be revised before copyediting and production can begin. Otherwise, you will be notified when your proofs are ready to be viewed.

Sincerely,
Siu-Kei Chow
Editor
Microbiology Spectrum